# WT1 Pulsed Human CD141+ Dendritic Cell Vaccine Has High Potential in Solid Tumor-Targeted Immunotherapy

**DOI:** 10.3390/ijms24021501

**Published:** 2023-01-12

**Authors:** Sung Yoon Cho, Seong Mun Jeong, Young Joo Jeon, Sun Ja Yang, Ju Eun Hwang, Byung Moo Yoo, Hyun Soo Kim

**Affiliations:** 1R&D Center, Pharmicell Co., Ltd., Seongnam 13229, Republic of Korea; 2Dr. Kim’s Stem Cell Clinic, 874 Eonju-ro Gangnam, Seoul 06027, Republic of Korea; 3Department of Gastroenterology, Ajou University School of Medicine, Worldcup-ro 164, Youngtong-gu, Suwon 16499, Republic of Korea

**Keywords:** DC vaccine, CD141, dendritic cells, zoledronate, T-cell activation, cancer immunotherapy, solid tumor, cancer antigens, Wilms’ tumor1 (WT1), tumor-associated antigens

## Abstract

Dendritic cells (DC) are powerful cells that play critical roles in anti-tumor immunity, and their use in cancer immunotherapy unlocks hidden capabilities as an effective therapeutic. In order to maximize the full potential of DC, we developed a DC vaccine named CellgramDC-WT1 (CDW). CDW was pulsed with WT1, an antigen commonly expressed in solid tumors, and induced with zoledronate to aid DC maturation. Although our previous study focused on using Rg3 as an inducer of DC maturation, problems with quality control and access led us to choose zoledronate as a better alternative. Furthermore, CDW secreted IL-12 and IFN-γ, which induced the differentiation of naïve T cells to active CD8+ T cells and elicited cytotoxic T lymphocyte (CTL) response against cancer cells with WT1 antigens. By confirming the identity and function of CDW, we believe CDW is an improved DC vaccine and holds promising potential in the field of cancer immunotherapy.

## 1. Introduction

Cancer is the leading cause of death across the world, and while traditional modes of treatments include surgery, chemotherapy, and irradiation, they often cause adverse side effects due to the inability to differentiate between cancerous and normal cells [1]. However, recent advancements in the field of immunotherapy allowed for the development of cancer vaccines, which aim to activate the body’s own immune system to specifically target cancer cells and consequently minimize side effects [2]. Cancer vaccines primarily use tumor-associated antigens or tumor-specific antigens to activate the antigen-specific lymphocytes of the immune system [3]. Activated lymphocytes, predominantly T cells, assume effector functions such as cytotoxicity and cytokine production to control cancer progression [4]. Different types of cancer vaccines utilize a specific set of immune cells, such as natural killer (NK) cells [5] and dendritic cells (DC) [6]. Of these, DC are antigen-presenting cells (APC) and play a critical role in activating the immune response via T cells. The main characteristic of DC involves the ability to capture antigens and process the protein into a peptide to be presented to T cells by major histocompatibility complex (MHC) molecules. However, DC comprise a heterogeneous population with each subset that carries distinct phenotypes and functions [7].

DC are divided largely into classical/conventional DC (cDC), plasmacytoid DC (pDC), and monocyte-derived DC (mo-DC). There are two broad groups of cDC: type 1 DC (cDC1), which primarily presents antigens using MHC class I to elicit CTL response from CD8+ T cell (CTL), and type 2 DC (cDC2), which uses MHC class II to promote a response from CD4+ T cell (helper T cell) [8]. Plasmacytoid DC are a unique subset of DC that are specialized in secreting type I interferon (IFN) [9]. Mo-DC are primarily involved in inflammation and promote TH17 immune response [10] (Figure 1A). Currently, mo-DC is most used in the field of DC anti-cancer immunotherapy research [11,12,13]. Although mo-DC is well-tolerated and safe, low therapeutic efficacy has hindered its widespread use. The limitation of mo-DC is demonstrated in vitro, where they show a limited ability to migrate to lymph nodes in order to activate strong cytotoxic T lymphocyte (CTL) responses [14]. In order to overcome the limitations of current modes of DC vaccine, cDC1 was chosen in this study, as it has the most superior antigen presentation ability and demonstrates high CTL response. Although studies on cDC1 vaccines are on the rise, cDC1 has not yet been explored in a clinical trial [15].

We obtained mononuclear cells (MNC) from bone marrow and then isolated CD34+ cells (hematopoietic stem cells) using the MACS^®^ Cell Separation technique. Cells were proliferated using granulocyte–macrophage colony-stimulating factor (GM-CSF), stem cell factor (SCF), and Fms-like tyrosine kinase receptor 3 (FLT3)-ligand, which directly induces differentiation into DC [16], and differentiation into DC was induced with GM-CSF and interleukin 4 (IL-4). Immature DC recognize WT1 protein as the antigen and are matured using zoledronate. WT1 (Wilms’ tumor1) antigen is highly expressed in various malignancies and a variety of solid tumors. Therefore, WT1 has been used as one of the targets of immunotherapy for cancer [17]. Zoledronate is a drug of the bisphosphonate class, which is widely used for the treatment of both osteoporosis and skeletal metastasis. Additionally, zoledronate inhibits the enzyme farnesyl diphosphate synthase, which plays a role in the mevalonate pathway and in the subsequent prenylation of small GTPase proteins, such as Ras [18]. The matured DC are then finalized as a DC vaccine (Figure 1A). In general, matured DC secrete various cytokines to induce the activation of immune cells [19] and bind directly to T cells for antigen presentation [20]. T cells reacted with DC become helper T cells (CD4+) to help the immune response or cytotoxic T cells (CD8+) to directly initiate anti-cancer effects [21] (Figure 1B). The composition of the DC vaccine was confirmed by flow cytometry, and the function of the vaccine was analyzed via cytokine secretion test, T cell change, and cytotoxic T Lymphocyte (CTL) assay.

## 2. Results

### 2.1. Flow Cytometry Profiles Illustrating DC Vaccine with High CD141+ Expression

To confirm the identity of the DC, various markers were analyzed, including CD141 (most widely used cDC1 marker), CD1c (marker of cDC2), and CD303a (marker of pDC). In addition, HLA-DR, CD80, and CD86, which are activation markers, were also analyzed. In contrast to 70% of cDC2 found in human DC of blood, the DC that was produced mostly consisted of cDC1 (CD141+ cells), and the double positive occurrence of CD141+CD1c+ in activated DC was also confirmed [22,23]. Furthermore, the results showed a significantly high level of activity for DC (Figure 2A,B). Cell morphology changes were also tracked using complete blood count (CBC) for different time points in DC production. While most of the cells proliferate as monocytes in the proliferation phase (Table 1), the differentiation phase resulted in a gradual decrease in monocytes to a level of disappearance toward the end of the production process.

### 2.2. Plasmatic Level of IL-12 and IFN-γ Cytokines Determined by ELISA

Of the many cytokines secreted by DC, the most representative are IL-12 and IFN-γ. IL-12 regulates inflammation by linking innate and adaptive immune responses. Most of the IL-12-induced effects are mediated by the secretion of IFN-γ and turned out to be critical for the induction of Th1 cells. IFN-γ plays a key role in the activation of cellular immunity and, subsequently, stimulation of antitumor immune response [24,25,26].

In order to verify the efficacy of CDW, IL-12 and IFN-γ secretion levels were analyzed via interaction with T cells. While T cell-only and T cell treated with unpulsed DC conditions resulted in secretion levels that were similar to each other, when the groups were compared with that of the T cell + CDW group, CDW induction on T cells yielded twice the level of IL-12(Figure 3A) and significantly increased the level of IFN-γ (Figure 3B).

### 2.3. Zoledronate’s Effect on the Differentiation and Maturation of cDC1 in CDW

In the previous study, Rg3 [27], a ginsenoside found in panax ginseng, was used to induce maturation of DC; however, the final product consisted of a layer of impurities, causing difficulties in quality control. Zoledronate was used as a substitute for Rg3 to overcome this problem, and its effect on the induction of DC maturation was analyzed. The most significant difference between the effects of the two substances is the ability of zoledronate to yield an exceptionally high expression of cDC1 (CD141+ cell) surface marker (Figure 4A). Thus, as cDC1 is known to be the most superior DC subtype in antigen presentation, zoledronate was chosen for the induction of DC maturation [18]. The optimal treatment time of zoledronate in inducing DC maturation was also assessed. Although 24 h treatment yielded a sufficient proportion of CD141+ cell, the CD141+ cell proportion in 3 h treatment result was greater by approximately 20%, and the CD86 (co-stimulatory marker) proportion in 3 h treatment was also greater by 30%. In addition, taking into consideration the mechanism of action of zoledronate, a shorter treatment time of 3 h compared to 24 h was deemed more effective in producing a higher quality of DC (Table 2).

### 2.4. CDW Vaccination-Induced WT1 Antigen-Specific T Cell Responses

Tests were performed to confirm the efficacy of the DC in activating T cells. T cells were isolated from peripheral blood mononuclear cells (PBMC), and T cell stimulators IL-2, Trans ACT, and DC were used for the primary induction. After seven days, secondary induction of DC was performed, and cells were recovered and investigated on the 10th day (Figure 5A). As a result, it was concluded that the CDW directly affected T cells given the increased number of CTL cells. In addition, a CTL assay was carried out to determine whether the CDW-educated T cells that were made also work in the WT1 antigen-expressing solid cancer model. After two induction of T cells with CDW, they were co-cultured with one of the following WT1-expressing cell lines: PANC1 (pancreatic cancer), MDA-MB-231 (breast cancer), Skov-3 (ovarian cancer), PC3, and Mia-paca2 (prostate cancer). Furthermore, in order to confirm whether CTL response was due to the effect of CDW on T cell activation, experiments using a T cell-only group activated by IL-2 and Trans ACT as well as a T cell group activated via unpulsed DC were carried out. While both these groups resulted in high killing response when in a 20:1 ratio, they failed to give concentration-dependent values in the setting of a 10:1 ratio (Figure 5C,D). In contrast, T cells induced by CDW group showed concentration-dependent action (Figure 5B).

### 2.5. Confirmation of the Safety of CellgramDC Vaccine

In both administration groups (3.4 × 10^4^ cells/animal or 1.7 × 10^5^ cells/animal), no abnormalities were observed with respect to death or general symptoms as a result of the administered substance.

During the observation period, no toxicologically significant changes were observed in the administration groups (3.4 × 10^4^ cells/animal or 1.7 × 10^5^ cells/animal) as a result of the administered substance. The observations included body weight (Figure 6A), urinalysis (Figure 6B), feed intake, ophthalmological examination, hematological examination, blood biochemical examination, organ weight, autopsy, and local tolerance test (Appendix A). Various tests were confirmed by comparing stem-derived DC (CellgramDC) and monocyte-derived DC (mo-DC). The survival and tumor size of mice were also tested. The tumor size in the stem-DC group was reduced by more than 50% compared to that of the mo-DC group, and the survival rate was also increased, confirming a strong anticancer effect (Appendix A).

We confirmed the stability and effectiveness of CellgramDC. CDW that is pulsed with WT1 and treated with zoledronate will also be tested for toxicity and efficacy and improved results are expected.

## 3. Discussion

In the CDW vaccine that was produced, cDC1 was the highest in proportion and showed a high level of activity. The cDC1 are cells with the most superior antigen presentation ability and are responsible for the main function of DC. Considering how cDC1 makes up a rare subset of DC [~0.03% of PBMC] [28], the predominance of the CD141+ population in CDW is a clear advantage in increasing the efficacy of the vaccine. Furthermore, cDC1 in CDW secreted cytokines (IL-12 and IFN-γ) at a high level and was capable of inducing the differentiation of naïve T cells to active CD8+ T cells. In our previous study, Rg3 was used as a maturation factor, but given the difficulty in obtaining supply and quality control of the final product, zoledronate was used as a substitute to overcome these challenges. DC induced by zoledronate was studied to analyze its role as an inducer of Vγ9 γδ T cell activation [29]. Zoledronate is a class of bisphosphonates, and bisphosphonates induce the activity of γδ T cells to rapidly and abundantly produce proinflammatory cytokines [30], and taking this into consideration, we designed our experiment to test the effects of short-term treatment. In our study, zoledronate (compared to Rg3) induced activation of CD8+ T cells in CDW and also increased the cDC1 level. The first clinical trial was conducted for DC induced by Rg3 (NCT 046158-45) from our previous study, but results have not been reported. However, we expect improved results in future clinical trials using DC induced by zoledronate given the current findings.

Recently, there are many studies surrounding immunotherapy against cancer, and the most researched therapies are CAR-T [31,32] and CAR-NK [33,34]. These therapies have confirmed their efficacy and are highly anticipated for the treatment of cancer. However, CAR-T is limited to the treatment of blood cancer, which comprises a very small proportion of all cancer types [35]. Furthermore, patients may suffer from side effects of the treatment, such as cytokine release syndrome (CRS) [36]. In order to overcome these shortcomings, immune therapy using NK cells is being researched. NK cells are highly potent lymphocytes and target cancer through multiple broadly expressed activating ligands. As a result, NK cells may address the limitations of autologous CAR T cell therapy. However, there are several potential drawbacks to the usage of NK cells, such as difficulty in cell culture, immediacy in the peak activity of cellular kinetics, and shorter intrinsic longevity as well as limited memory phenotype in the life span and response [34,37,38]. Other research involves combinational therapies using DC or T cells and immune checkpoint inhibitors such as anti-PD1 (programmed cell death protein1) [39] or anti-PD-L1 (programmed death-ligand1) [40]. The co-administration of these drugs allows the targeting of the immunosuppressive tumor microenvironment and further research is in progress to increase the efficacy of these therapies.

The research for the development of DC vaccine for immunotherapy is continually increasing, and many advancements are being made in this field. DC plays a central and critical role in the advanced immune system, and the DC vaccine may offer an advantage compared to other modes of immunotherapy for cancer. Because DC does not directly kill cancer cells, normal cells are unaffected, eliminating adverse side effects. Furthermore, by pulsing DC with WT1, an antigen commonly expressed in many solid tumors [17], the DC vaccine can target solid tumor models. Before deciding on the WT1 protein antigen, we experimented with various types of antigens. By using quality guaranteed products, as well as various antigens such as a peptide, peptivator, or pepmix, the antigen with the greatest effect was when used as a protein. Consequently, we used WT1 protein for antigen pulsing. The most widely studied approach in DC therapy uses mo-DC pulsed with WT1 in conjunction with chemotherapy [12]. The safety and immunogenicity of mo-DC has been confirmed through clinical trials [13]. While other DC vaccine research develops its vaccines using monocyte-derived DC from blood, we predicted that DC derived from stem cells will have increased potential. Additionally, it is known that cDC has greater power than mo-DC, propelling us toward this research. Therefore, CDW may be a better alternative to mo-DC, as its primary component is cDC1, a DC subtype with the most effective cross-presentation ability. Although the cytotoxic activity was similar in the T cells+unpulsed DC group and CDW group, we hypothesize this is because unpulsed DC is also a type of cDC1 cell. Regardless, CDW may effectively elicit a strong anti-tumor immune response by increasing the cDC1 population. Through the preclinical study, we tested repeated dose toxicity of the DC vaccine. The development and production process of CDW have been verified, and we hope to conduct further studies to test the improved effects of CDW. While efficacy was confirmed in vitro, a dose toxicity study was performed for efficacy validation in vivo.

Once the results of clinical trials using CDW are reported, we intend to guide our research in the direction that will improve current research.

## 4. Materials and Methods

### 4.1. Cell Culture

DC culture is examined. Human bone marrow-derived mononuclear cells (MNC) were obtained from Lonza (2M-125C). Hematopoietic stem cells were isolated from MNC by magnetic separation, using the CD34 microbead kit Ultrapure human (Miltenyibiotec, Bergisch Gladbach, Germany 130-100-453). GM-CSF (PeproTech, Cranbury, NJ, USA, AF-300-03-1000), SCF (PeproTech, AF-300-07-1000), and FLT-3ligand (PeproTech, AF-300-19-1000) were used for proliferation, lasting about 14 days. Thereafter, GM-CSF and IL-4 (PeproTech, AF-200-04-1000) were used to differentiate cells into DC for one week. At the end of the differentiation process, DC was pulsed with WT1 protein for antigen recognition and maturation was induced with zoledronate. After maturation, CDW is complete.

T cell culture is examined. Human peripheral blood mononuclear cells (PBMC) were obtained from Lonza (CC-2702). Naïve T cells were isolated from PBMC by magnetic separation using the Pan T cell isolation kit human (Miltenyibiotec, 130-096-535). IL-2 (PeproTech, 200-02) was used for proliferation, lasting about 10~14 days. TransAct (Miltenyibiotec, 130-111-160) and CDW were used for activation on the first and seventh days of culture.

These cells were routinely grown in HyClone RPMI 1640 media (Sigma-Aldrich, Burlington, MA, USA, SH30255.01) supplemented with 10% Fetal bovine serum (Gibco, Carlsbad, CA, USA, 10099-141) and 1% Gentamicin (Gibco, 15710-064) at 37 °C in a humidified atmosphere containing 5% CO_2_.

### 4.2. CTL Assay

The Cell Counting Kit-8 (CCK-8, Dojindo Laboratories, Kumamoto, Japan) was used to measure the cytotoxicity of T cells in cancer cells. The cancer cells (1 × 10^4^ cells/well) were cultured with T cells (1 × 10^4^ cells/well, 1 × 10^5^ cells/well, 2 × 10^5^ cells/well) and induced twice with CDW for 72 h in 96-well plates at 37 °C in a 5% CO_2_ incubator. After 10 μL CCK-8 solution was added to each well, the plate was re-incubated for 3 h at 37 °C, and the absorbance at 450 nm was detected using a microplate reader (Epoch™ Microplate Spectrophotometer, BioTek U.S, Winooski, VT, USA). Data were analyzed using SigmaPlot (Systat, version 8.02a) software.

Cancer cells used were: PANC1 (CRL1469), MDA-MB-231 (HTB-26), Skov-3 (HTB-77), PC3 (CRL-1435), and Mia-paca2 (CRL1420). All cancer cells were purchased from ATCC.

### 4.3. Phenotypic Analysis

Flow cytometry was used to analyze the DC and T cell phenotype. For analysis, DC was stained with CD80 (AF700, BD, 56113), CD86 (PE, 12-0869-42), HLA-DR (PE, 12-9956-42), CD1c (PC5.5, 46-0015-42), CD141 (APC, 17-1419-42), and CD303a (FITC, 11-9818-42). T cells were stained with Fixable viability dye (AF750, 65-0865-18), CD3 (PE, 12-0038-42), CD4 (FITC, 11-0049-42), CD8a (PerCP-eFF710, 46-0087-42) and CD45RO (APC, 17-0457-42). Data were acquired on a Cytoflex cytometer (Beckman Brea, CA, USA) and analyzed using FlowJo (Treestar, version 10.8.1) software. Cell morphology changes were tracked using a complete blood count (CBC) (Beckman). All antibodies except CD80 were purchased from Invitrogen.

### 4.4. ELISA Assay of Cytokine Secretion

The concentration of cytokines was measured using commercial Quantikine ELISA Kits (R&D systems, Bio-Techne) to detect IFN-γ (DIF50C) and IL-12p70 (D1200), which are secreted in T cell and T cell + CDW co-culture supernatant, and the absorbance at 450 nm was detected using a microplate reader (Epoch™ Microplate Spectrophotometer, BioTek U.S).

### 4.5. Animals

Pathogen-free female C57BL/6 mice at 6 weeks old were purchased from Orient Bio (Seongnam, South Korea). The test was performed at Biotoxtech Co.Ltd (Ochang-eup, Cheongju-si, Chungcheongbuk-do, 28115, South Korea). C57BL/6 mice were used in 1-week intervals for a total of 6 times. Cells were administered subcutaneously near the draining lymph node. The amounts were low dose (normal dose × 10), middle dose (normal dose × 50), and high dose (normal dose × 100). Each of these dosage amounts are as follows: low dose (3 × 10^5^), middle dose (1.5 × 10^6^), and high dose (3 × 10^6^).

Normal dose = 1 × 10^7^/60 kg vs. mouse 20 g.

Mice were housed at a pathogen-free animal care facility and kept on diurnal cycles of 12 h light and 12 h dark with ad libitum access to food and water. Animal care was performed following the OECD Principles of Good Laboratory Practice guideline. Mice were acclimated for at least one week before any experiments were conducted.

Ethics committee name: Association for Assessment and Accreditation of Laboratory Animal Care International (AAALAC).

Approval Code: 180159. Approval Date: 2018.03.06.

### 4.6. Statistical Analysis

Comparisons of samples for the establishment of statistical significance were determined by two-tailed student’s *t*-test. Results were considered statistically significant when * *p* < 0.05, ** *p* < 0.01 and *** *p* < 0.001

### 4.7. Reagents

Ginsenoside Rg3 was supplied by Dr. Sung Ho Son (VitroSys Inc., Yeongju, Republic of Korea).

Zoledronate (zoledronic acid), kindly provided by Novartis Pharma AG, was added on the last day of culture to evaluate its immuno-modulatory effects on DC. The drug concentration was selected on the basis of a series of tests performed that determined the ideal concentration of zoledronate to be 1 μmol/L.

Human Wilms’ tumor gene 1 (WT1) protein was synthesized at JW CreaGene (Seongnam, Republic of Korea). The purity of the protein was confirmed to be >95% by sodium dodecyl sulfate–polyacrylamide gel electrophoresis (SDS-PAGE). Synthetic protein was dissolved in dimethyl sulfoxide (DMSO) according to the manufacturer’s recommendations and stored at −70 °C until use.

## Figures and Tables

**Figure 1 ijms-24-01501-f001:**
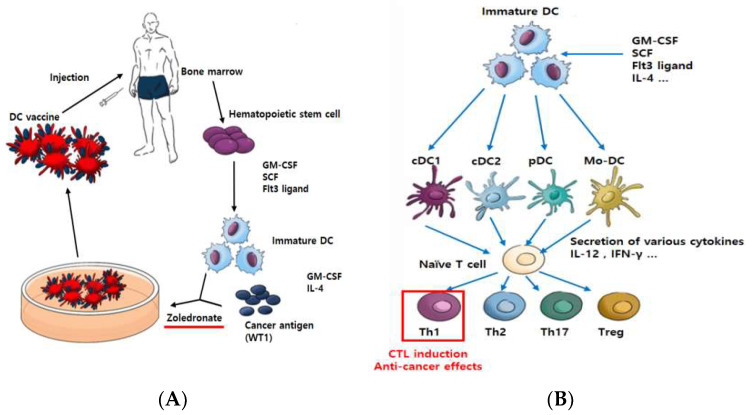
Scheme of vaccination with ex vivo pulsed-DC (against cancer). Dendritic cells play an important role in deciding the direction of host immune reactions, CTL induction, and anti-cancer effects. CellgramDC-WT1 (CDW), a Th1-inducing type of dendritic cell vaccine, is made as follows. Hematopoietic stem cells (CD34+ cells) are isolated from the patient’s bone marrow for proliferation and differentiation into DC. Then, the cells are pulsed using WT1 protein and made into a DC vaccine using zoledronate as a maturation factor (**A**). Through different growth factors, such as GM-CSF and FLT3-ligand, immature DC differentiate into several types of DC. These differentiated DC secrete various cytokines to regulate the immune response. Specifically, naïve T cell differentiation into Th1-type T cells can occur, which directly induces CTL reactions to cause anti-cancer effects (**B**).

**Figure 2 ijms-24-01501-f002:**
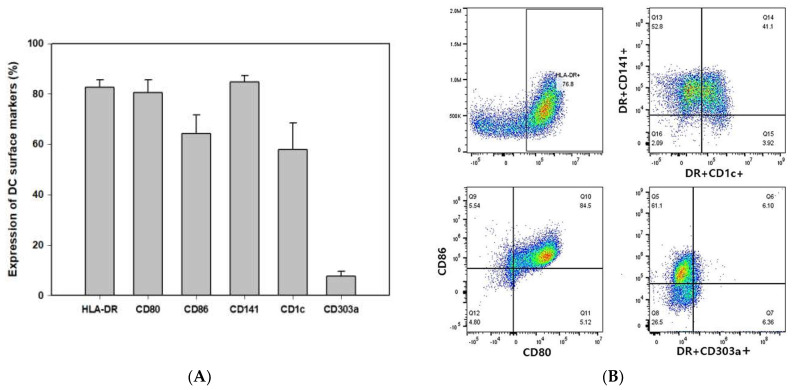
Identification of CDW subsets. Phenotypic characteristics of DC. During the differentiation process, the DC were pulsed with WT1 protein and treated with 1 μM zoledronate for 3 h. The data show the expression of stimulatory marker and subtype of DC representative of human DC (n = 5) (**A**). Results are shown as dot plots (**B**).

**Figure 3 ijms-24-01501-f003:**
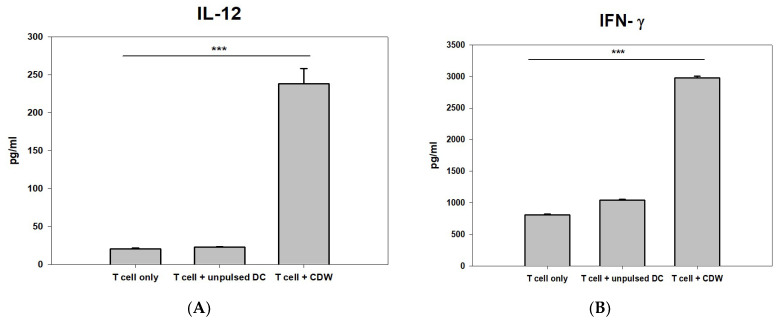
Induction comparison of CDW on T cells via cytokine analysis. The secretion of IL-12 (**A**) and IFN-γ (**B**) was measured in T cell only (activated IL-2 and Trans ACT), T cell + unpulsed DC, and T cell + CDW co-culture supernatant using ELISA assay (n = 3). ELISA was performed using the supernatant at the time of completion. Analysis was performed through SigmaPlot. *** *p* < 0.001.

**Figure 4 ijms-24-01501-f004:**
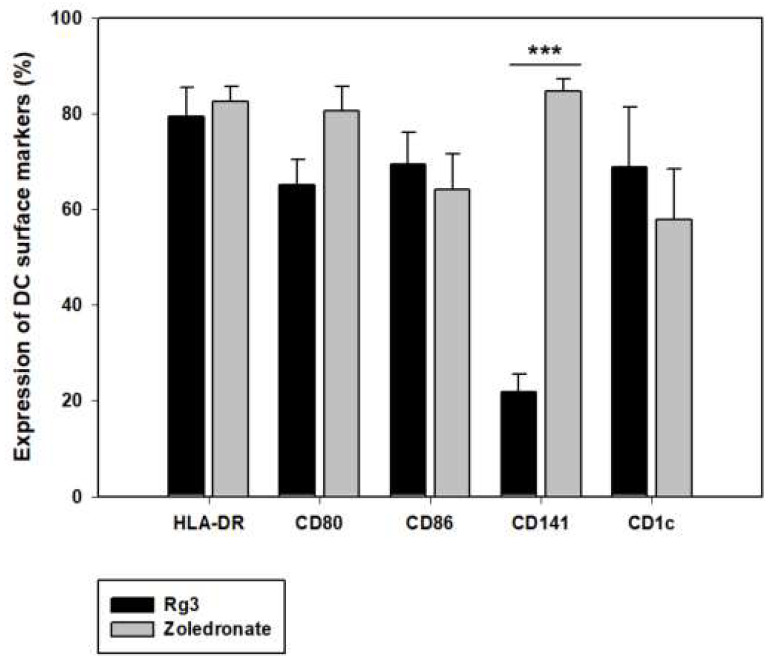
Effect of zoledronate on the differentiation and maturation of cDC1 in DC vaccine production. Effects of zoledronate in DC vaccine. In the process of DC vaccine production, 3 hr treatment with zoledronate induces differentiation and maturation of DC to cDC1 and yields a higher level of CD141 marker. Phenotype markers were analyzed by flow cytometry to compare Rg3 (n = 4) and zoledronate (n = 5), which were used for the induction of DC maturation. *** *p* < 0.001.

**Figure 5 ijms-24-01501-f005:**
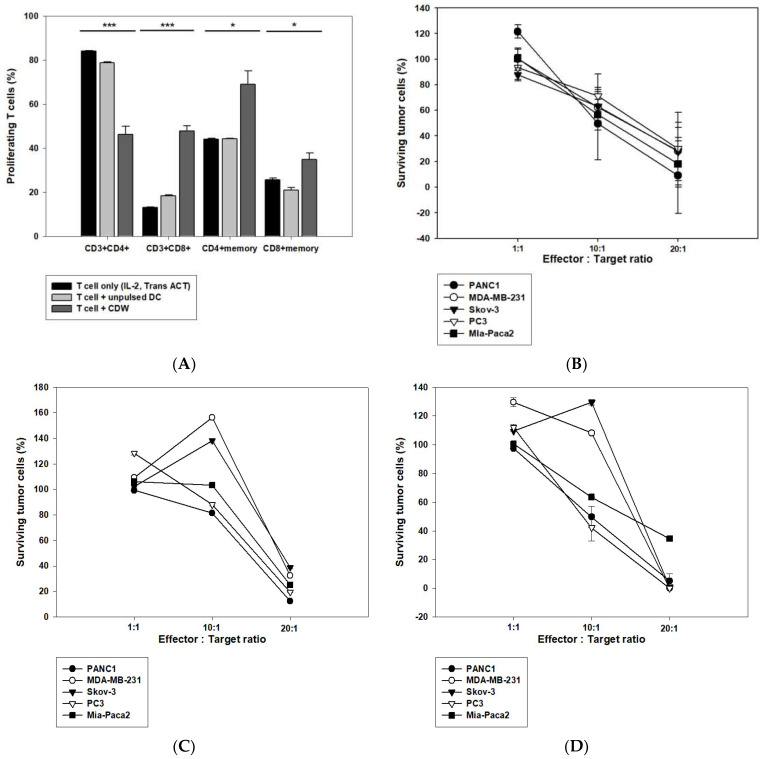
CDW increases CD8+ T cells to promote cytotoxicity against cancer cells. Effect of CDW on T cell response assessed via CTL. IL-2 and Trans-Act are T cell stimulators and were used to stimulate T cells. The activated T cells were co-cultured with DC for the first induction, which lasts for seven days and the second induction which extends to 10 days. The changes in the T cell subtype were analyzed via flow cytometry (**A**). The T cells cultured for 10–14 days were co-cultured with cancer cells expressing WT1 according to appropriate ratios in a 96-well plate. Post-72 h, the survival rate of cancer cells was analyzed using CCK8 (**B**–**D**). T cells induced by CDW group (**B**). T cell only group (activated IL-2 and Trans ACT) (**C**). T cells induced by unpulsed DC group (**D**). * *p* < 0.05, *** *p* < 0.001.

**Figure 6 ijms-24-01501-f006:**
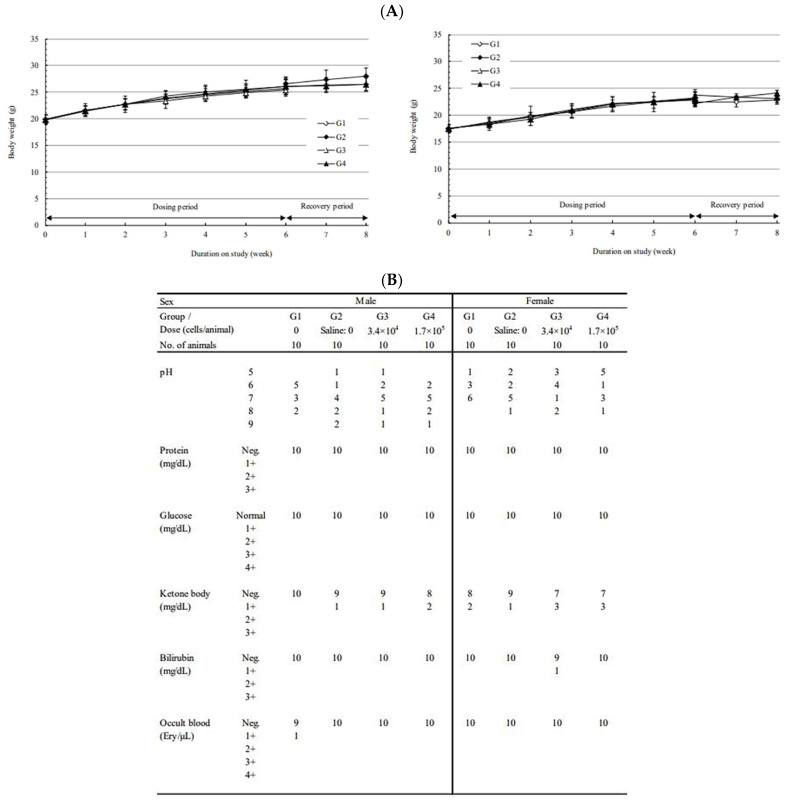
Subcutaneous dose toxicity study of CellgramDC in C57BL/6 mice. To test the safety and toxic response of the CellgramDC, female and male mice of C57BL/6 strain were subcutaneously injected with CellgramDC for a total of six weeks (one injection per week). The safety of CellgramDC was tested by subcutaneous injection into female and male mice for a total of six weeks (one injection per week). Administration groups consisted of two groups: 10 mice injected with 3.4 × 10^4^ cells/animal and 15 mice injected with 1.7 × 10^5^ cells/animal. A negative control group was comprised of 15 mice and were injected intravenously with a solution composed of excipient, plasma solution-A/human serum albumin (HSA) 90% + DMSO 10% and saline for six weeks (one injection per week). In order to test for a reversible toxic response, five mice from each comparison group, negative control group, and 1.7 × 10^5^ cells/animal administration group, were given two weeks of the recovery period. During the recovery period, weight check (**A**), urinalysis (**B**), general symptoms, feed intake measurement, and ophthalmological examination were observed. Following the observation period, hematological tests, blood biochemical tests, and organ weight measurements were performed, as well as visual and histopathological examinations at necropsy.

**Table 1 ijms-24-01501-t001:** During the process of proliferation and differentiation from CD34+ cells to DC, phenotypic changes were analyzed using CBC.

Days	Lymphocyte (%)	Monocyte (%)	Neutrophils (%)
0	60.17	32.45	7.12
4	14.89	44.69	39.36
7	5.76	69.78	23.02
11	12.33	71.24	14.38
14	14.53	76.92	5.13
18	6.69	91.74	1.18

**Table 2 ijms-24-01501-t002:** Effect of zoledronate treatment times (3 h and 24 h) on surface markers.

%	HLA-DR	CD80	CD86	CD141	CD1c
Zol 24 h (n = 3)	76.27	84.67	36.07	64.77	80.97
Zol 3 h (n = 3)	82.62	80.54	64.23	84.68	57.94

## Data Availability

Data is contained within the article or Appendix A. The data presented in this study are available in this manuscript.

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
