# Peer review of "WT1 Pulsed Human CD141+ Dendritic Cell Vaccine Has High Potential in Solid Tumor-Targeted Immunotherapy"

_ijms, 2023, doi:10.3390/ijms24021501_

Round 1
Reviewer 1 Report
Manuscript by Sung Yoon Cho and co-authors contains data concerning preclinical evaluation of safety and efficacy of DC vaccine pulsed with WT1 peptide. The safety of this vaccine is supported by the data provided (although DC vaccines are rather safe in general), but the efficacy of this vaccine remains unknown, due to the poor design of experiments:
fig 3 – there is no control of coincubation of T-cells with non-pulsed dendritic cells (or pulsed with irrelevant peptide). Therefore the effect of specific T-cell activation could not be differentiated from non-specific activation by DC-secreted cytokines.
Fig 5a – control with T-cells activated with IL-2 and TransACT, but without DC is missing. IL-2+TransACT is very powerful combination that could easily drive T-cells towards phenotypic changes observed in the experiment (without any help from DC). DC effect on T-cells should be proven.
Fig 5b – No control with T-cells, that were activated with IL-2+TransACT only. There are no evidences, that observed T-cell cytotoxicity is DC-driven.
Author Response
Thank you for taking the time to review our article. We appreciate your comments and have provided responses to each of them as well as appropriate revisions.

Reviewer 2 Report
The field of cancer immunotherapeutic is rapidly growing and this experimental article addresses preclinical steps toward cancer vaccine development.
Several issues however preclude publication in the present form:
1. The introduction skip crucial points:
a. Identity, function an reasons to chose WT1 protein as antigen for the development of a cancer vaccine and types of cancers intended to threat with such therapy
b. The used of zoledronate as an adjuvant for vaccine development was remarked in the abstract but not discussed. From the paper “In addition, taking into consideration the mechanism of action of zoledronate, a shorter treatment time of 3 hr compared to 24 hr was deemed more effective in producing a higher quality DC (Figure 4. B)” unfortunately we have to read reference 24 to know about zoledronate.
c. Is it usual to use human antigen pulsed DC in mice immunization? In any case could you move the discussion on the current status of cancer vaccines from the discussion section to the introduction?
2. Materials and methods are deficient/obscure:
a. The methods for cDC are not referenced (ref15 is a review)
b. In order to verify the efficacy of CDW, IL-12 and IFN-γ secretion levels were analyzed via interaction with T cells. How can you analyze cytokine by interaction with T cells? This in a chapter named “Plasmatic level of IL-12 and IFN-γ cytokines determined by ELISA”
c. Zoledronate 3 hours cause immediate DC differentiation?
d. After 7 days, secondary induction of DC was performed, and cells were recovered and investigated on the 10th day. This is the central experiment in the paper but that’s all about the protocol.
e. Crucial information about the amount of cells used for in vivo experiments, way of administration, amount of antigen used and so on are missing. How will this compare to in human experiments?
3. Control missing
a. Could you test a WT1 negative cell in Fig 5B CTL assay?
b. Have you got authorization to animal studies from an ethical committee?
4. Discussion often not supported by the data nor by references.
a. Considering how cDC1 makes up a rare subset of DC [~0.03% of PBMC] [25], the predominance of the CD141+ population in CDW is a clear advantage in increasing the efficacy of the vaccine. This probably refer to some study (not discussed) demonstrating cDC are better for cancer vaccination.
b. The first clinical trial was conducted for DC induced by Rg3 from our previous study but results have not been reported. Could you at least indicate some trial number.
c. Discuss the relevance of your finding for vaccine development not the status of the field.
d. Once the results of clinical trials using CDW are reported, we intend to guide our research in the direction that will improve current research. Is this a scientific paper or a Christmas letter?
As I’m not an expert in the field, those problems makes the article hard to follow and difficult to appreciate. English is fluent per se but a full re-writing is required.
Minor points
· 98 - Furthermore, the activity of the DC reported at a significantly high level (Figure 2. A, B). OBSCURE MEANING
· 294 Furthermore, cDC1 in CDW secreted cytokines (IL-12 and IFN-γ) at a high level and was capable of inducing the differentiation of naïve T cells to active CD8+ T cells. OBSCURE MEANING
· 99 Cell morphology changes were also tracked using complete blood count (CBC) for different time points in DC production. While most of the cells proliferate as monocytes in the proliferation phase, the differentiation phase resulted in a gradual decrease of monocytes to a level of disappearance toward the end of the production process (data not shown). THIS IS IN CONTRAST WITH TABLE 1 ???
· Fig 1 as the following is a strange figure: it contains a table (also others) and 1B is nearly unreadable (also others)
Author Response

(The authors gave the same response as above.)

Reviewer 3 Report
Dr. Hyun Soo Kim et al. did a vitro experiment to evaluate the immune response to WT1 pulsed CD141+ DC vaccine. Authors observed increased IL-12 and IFN-gamma in T cell + CDW group, but not in T cell only or T cell + unpulsed DC groups. In addition, CD8+ and CD4+memory T cells were increased after CDW treatment. Furthermore, the survival rates of cancer cells were reduced in the setting of a 20:1 ratio in several tumor cell types. At last, authors confirmed the safety of CDW by testing body weight, urinalysis and et al. However, there are some concerns.
1. Lack of a description of current CD141+ vaccine studies in the introduction.
2. Were there any significant improvements of killing tumors using CDW compared to T cell only and T cell + unpulsed DC groups?
3. Is that possible to evaluate the CD8+ response or IFN-gamma level after CDW treatment in vivo.
Author Response
Hello,
We appreciate you taking the time to review our manuscript. Thank you for your feedback, and we hope our revised manuscript reflects the necessary changes.
Regards,
Esther Hwang

Reviewer 4 Report
The study investigates the efficacy of WT1-pulsed, zoledronate-matured human dendritic cells in inducing cytotoxic T cells response for WT1-expressing solid cancers. Specific comments are listed below:
(1) The introduction about WT1 antigen and the nature of zoledronate is lacking. The information covered in the introduction section is too shallow and superficial. The same was seen in the discussion section as well.
(2) You mentioned that classical DC1 cells primarily present antigens using MHC I to elicit CD8+ CTL response. In the study, you claimed that the pulsed and matured DC cells produced are mainly cDC1 type and the cell surface markers in use include HLA-DR which turned out to be 76.27% and 82.62% in 24hrs and 3hrs zoledronate induction. But HLA-DR is class II MHC.
(3) How sustainable is the secretion of IL-12 and IFN-γ in co-culture? What time point the ELISA was done in Figure 3?
(4) Did you validate activated T cells' phenotype (surface markers) after co-culture with CDW?
(5) Figure presentation must be improved. Many labels are not aligned well. And some figures are cut short e.g. top two figures in Figure 5.
(6) English editing is needed.
Author Response

(The authors gave the same response as above.)

Round 2
Reviewer 1 Report
Figures from supl1 should be moved to the main body of the manuscript (e.g. 5c, 5d), since these figures illustrates efficacy of the product. Statistical analysis of the results (significance of the difference between cytotoxicity of T-cells only group and other groups) should be performed.
Passage:
"Thus, CDW may effectively elicit a strong anti-tumor immune response by
increasing the cDC1 population. Through the preclinical study, we tested the safety,
repeated dose toxicity, and efficacy of the DC vaccine. The development and production
process of CDW have been verified, and we hope to conduct further studies to test the
improved effects of CDW."
should be tuned down, since authors do not shown "strong anti-tumor responce".
For most of the cancer cell lines tested T-cells+unpulsed DC showed rather similar cytotoxic activity. This fact should be discussed.
Author Response
Thank you for taking the time to review our manuscript. We have made changes according to your comments, and hopefully, they are reflected in the manuscript.

Reviewer 2 Report
None of the concerns/indication was taken up in the text. Many interesting responses were present in the rebuttal letter but not used to ameliorate the manuscript.
In addition note that conflict of interest statements are missing and such conflicts are apparently affecting the manuscript from rebuttal letter.
Author Response

(The authors gave the same response as above.)

Reviewer 3 Report
My concerns were addressed. Thank you.
Round 3
Reviewer 1 Report
Two minor points:
Since all experiments concerning efficacy were performed in vitro, phrases such as
"We hypothesized that enhanced DC maturation via zoledronate can
modulate the tumor microenvironment and immune evasion factors. To confirm this hypothesis,
the effect of zoledronate on DC maturation was tested in vitro"
should be restructured to be more accurate. Modulation of TME by zoledronate was not directly studied in this research.
Also statement in the end of discussion
"Through the preclinical study, we tested the safety,
repeated dose toxicity, and efficacy of the DC vaccine."
Should be re-written, because typical preclinical evaluation of efficacy inculde in vivo efficacy validation. It should be clear for all readers, that only in vitro efficacy evaluation was done.
Author Response

(The authors gave the same response as above.)

Reviewer 2 Report
This new version of the article satisfies the requirements for publication.
Author Response
Hello,
We appreciate you taking the time to review our manuscript.
Regards,
Esther Hwang